# Assessment of Fatty Acid Desaturase (Fads2) Structure-Function Properties in Fish in the Context of Environmental Adaptations and as a Target for Genetic Engineering

**DOI:** 10.3390/biom10020206

**Published:** 2020-01-31

**Authors:** Zuzana Bláhová, Thomas Nelson Harvey, Martin Pšenička, Jan Mráz

**Affiliations:** 1South Bohemian Research Center of Aquaculture and Biodiversity of Hydrocenoses, Faculty of Fisheries and Protection of Waters, University of South Bohemia in České Budějovice, Zátiší 728/II, 389 25 Vodňany, Czech Republic; 2Centre for Integrative Genetics (CIGENE), Department of Animal and Aquacultural Sciences, Faculty of Biosciences, Norwegian University of Life Sciences, 1430 Ås, Norway

**Keywords:** fatty acyl desaturase, Δ6 - desaturase, long-chain polyunsaturated fatty acid, LC-PUFA, ω3, ω6, EPA, DHA, AA, essential fatty acid, health, fish, transgene

## Abstract

Fatty acid desaturase 2 (Fads2) is the key enzyme of long-chain polyunsaturated fatty acid (LC-PUFA) biosynthesis. Endogenous production of these biomolecules in vertebrates, if present, is insufficient to meet demand. Hence, LC-PUFA are considered as conditionally essential. At present, however, LC-PUFA are globally limited nutrients due to anthropogenic factors. Research attention has therefore been paid to finding ways to maximize endogenous LC-PUFA production, especially in production species, whereby deeper knowledge on molecular mechanisms of enzymatic steps involved is being generated. This review first briefly informs about the milestones in the history of LC-PUFA essentiality exploration before it focuses on the main aim—to highlight the fascinating Fads2 potential to play roles fundamental to adaptation to novel environmental conditions. Investigations are summarized to elucidate on the evolutionary history of fish Fads2, providing an explanation for the remarkable plasticity of this enzyme in fish. Furthermore, structural implications of Fads2 substrate specificity are discussed and some relevant studies performed on organisms other than fish are mentioned in cases when such studies have to date not been conducted on fish models. The importance of Fads2 in the context of growing aquaculture demand and dwindling LC-PUFA supply is depicted and a few remedies in the form of genetic engineering to improve endogenous production of these biomolecules are outlined.

## 1. Introduction

Fatty acid desaturase 2 (Fads2) is an endoplasmic reticulum membrane bound protein which acts as the first enzyme in the biosynthesis of long chain (≥ C_20_) polyunsaturated fatty acids (LC-PUFA). This pathway includes physiologically important eicosapentaenoic acid (EPA, ω3-20:5^5,8,11,14,17^), docosahexaenoic acid (DHA, ω3-22:6^4,7,10,13,16,19^), and arachidonic acid (AA, ω6-20:4^5,8,11,14^) which are produced from the shorter and lower level polyunsaturated fatty acids (PUFA) α-linolenic acid (ALA, ω3-18:3^6,9,12^) and linoleic acid (LA, ω6-18:2^9,12^). Human and many fish genomes encode for Fads2 as well as for some other enzymes acting in the LC-PUFA biosynthetic pathway, namely Fads1, elongase 5 (Elovl5), elongase 4 (Elovl4) or elongase 2 (Elovl2). LC-PUFA are often referred as conditionally essential nutrients, meaning that however the organism could be capable to produce them, this endogenous production is insufficient to meet demand, hence, LC-PUFA biomolecules must be obtained through the diet. Endogenous production is hypothesised to serve as a compensation apparatus, which helps the organisms to maintain homeostasis under fluctuating environmental conditions and LC-PUFA availability. This could be the reason why marine fish, unlike freshwater fish, do not have the capability to produce LC-PUFA at a significant level [1] as a consequence of living in nutritionally rich oceans. In contrast to the conditional-essentiality of LC-PUFA, precursors of LC-PUFA, LA and ALA, cannot be created de novo in nearly any living animal, since their genomes do not encode for enzymes capable to create them (such would be methyl-end desaturases with Δ12 and Δ15 activity converting oleic acid (18:1n-9) into LA (18:2n-6) and ALA (18:3n-3)) [2]. Hence, animals are usually dependent on plants for providing double bonds in the Δ12 and Δ15 positions of the two major precursors of the ω6 and ω3 fatty acids LA and ALA [3]. These two fatty acids, therefore, are called essential fatty acids. In the literature, however, most often, the conditional-essentiality of AA, EPA and DHA for vertebrates and humans is not considered and with LA and ALA, these biomolecules are altogether called essential fatty acids (EFA).

The exogenous supply of EFA for many animals, including some omnivorous terrestrial animals and humans, is from aquatic ecosystems. In aquatic ecosystems, substantial amounts of EPA and DHA are provided by primary producers. Historically, the primary production of these biomolecules has been associated exclusively to single-cell microorganisms such as photosynthetic microalgae, heterotrophic protists and bacteria. Recently, however, multiple invertebrates, many of them representing abundant groups in aquatic ecosystems, have been confirmed to be able to produce PUFA de novo and farther biosynthesize them into ω3 LC-PUFA similarly to single-cell microorganisms [4,5]. Once synthesized by microalgae or invertebrates, these biomolecules are transferred through trophic webs to organisms of higher trophic levels. Fish are considered as the best source of ω3 LC-PUFA for humans. However, anthropogenic factors such as pollution, eutrophication, climate change or biological invasions threaten the LC-PUFA production by primary producers at present. World capture fishery production cannot be increased and aquaculture is expected to be continuously growing to deliver food to humans. Here, this could be seen a paradox. While aquaculture has increasingly become the major source of EPA and DHA for humans, it has also, at the same time, become the greatest consumer of the world’s available supply of EPA and DHA. The problem of bridging the gap between supply and demand of LC-PUFA was very recently excellently reviewed by Tocher et al. [1]. Since LC-PUFA have been identified as globally limited nutrients, the ability of an organism to compensate for dietary deficiencies of LC-PUFA by enhanced activity of its endogenous biosynthesis is of great importance for human and animal health as well as for the maintenance of fish as an EFA source for the human diet and aquaculture food.

Although it has been confirmed that alterations in activities of elongases Elovl4 and Elovl5 catalysing subsequent steps in LC-PUFA biosynthesis can alter EPA, DHA and AA production by promoting various disease states [6], Fads2 is still commonly considered as the rate-limiting and the most important enzyme of LC-PUFA biosynthesis. Moreover, in light of some recent significant publications, Fads2 appears as an enzyme with far-reaching implications for environmental sustainability.

## 2. Significance and Essentiality of LC-PUFA Biomolecules

LC-PUFA are important components of fat and in higher eukaryotes confer fluidity, flexibility and selective permeability to cellular membranes. They greatly influence many physiological processes. Participation of LC-PUFA in several major human pathologies (inflammatory-autoimmune diseases, cardiovascular diseases, cancer and neurodegenerative disorders) has been reviewed recently by Zárate et al. [7]. The pivotal role of lipids as an essential dietary component is now widely accepted; however, many decades of research have gone into this conclusion, which has recently been very well reviewed by Spector and Kim [8]. Fish played an essential role in coming to this conclusion. The essentiality of any fatty acid biomolecule was, for the very first time, reported in what was at the time highly controversial scientific work of George Oswald Burr in 1929 [9,10]. He and co-workers demonstrated that LA and ALA rescued the growth retardance phenotype in rats fed fat-free diet and found the first clue that LA is a precursor of AA. The concept of essential fatty acids appeared [11]. But it was before chain desaturation or elongation of fatty acids had been demonstrated and the authors wondered how two double bonds + two double bonds could equal four [12], as was elucidated later in series of studies conducted by Mead et al. [13]. The linkage between LA, its ω6 fatty acid desaturation and elongation products, and the formation of prostaglandins as biomediators was reported by Bergström et al. [14,15], who stated“…* the symptoms of essential fatty acid deficiency at least partly are due to an inadequate biosynthesis of the various members of the prostaglandin hormone system.*” The pathway through which ALA is converted into EPA and DHA was determined in 1960 by Klenk et al. [16]. Noteworthily, no important functions were attributed to ω3 fatty acids for a long time. However, this changed in 1968 when Dr. Jørn Dyerberg made the remarkable discovery that fats in the diet of Greenland Eskimos comprised mainly of fish were associated with a lowered risk of cardiovascular diseases, concretely, that plasma of these people contained large amount of ω3 fatty acids and their phospholipids contained high levels of EPA, but very little AA [17]. Their conclusion was that EPA protects against cardiovascular diseases [18]. Without a doubt, the paradigm had been changed and fatty acids were no more considered to fulfil the only function in energy storage. Widespread interest was awoken in investigations of unsaturated fatty acids as biomolecules indispensable for health. Numerous global and national health agencies and associations and government bodies have produced many recommendations for EFA intake for a healthy human diet through fish consumption. With the advent of molecular and genetic technologies, there appeared much evidence that a balanced abundance of EFA is a prerequisite for health and disease prevention in humans [19,20,21]. Meeting the dietary demands of a burgeoning human population with a correct dietary balance of EFA and at levels required for normal health and development has become a major challenge. It has been clear that understanding the molecular basis of LC-PUFA biosynthesis would underpin efforts to meet this challenge. Various strategies of human populations regarding EFA metabolism have been shown by Gladyshev and Sushchik this year [22]. Studies performed in fish are advantageous mainly because there is wide variation between fish species in their ability to biosynthesize LC-PUFA, probably as a consequence of inhabiting widely different environments. Comparisons of their genomes and expression levels of genes encoding key elements in the LC-PUFA biosynthetic pathway between species have been promising to increase knowledge of the molecular components of the pathway and of the molecular genetic basis of phenotypic variation in LC-PUFA biosynthesis.

## 3. Fads2 in LC-PUFA Biosynthesis

The biosynthesis of C_20-22_ LC-PUFA involves alternating steps of desaturation (introduction of an additional double bond) and elongation (addition of two carbons) of the dietary essential C_18_ fatty acids LA and ALA [1]. Firstly, in the biosynthesis of LC-PUFA, Δ6 Fads2 desaturase converts dietary obtained LA (18:2n-6) and ALA (18:3n-3) into gamma-linoleic acid (GLA) and stearidonic acid, respectively. Subsequently, in the biosynthesis of LC-PUFAs of ω3 series, it converts tetracosapentaenoic acid into tetracosahexaenoic acid which is then converted to DHA. Enzymatic steps in the biosynthesis of LC-PUFAs in vertebrates are shown in Figure 1 [23,24,25,26,27,28,29,30]. AA and EPA are biosynthesized in the same pathway in which LA and ALA substrates compete of the same enzymes, respectively. The pathway revealed from studies in vertebrates are the so-called “Δ6 pathway” (Δ6 desaturation–elongation–Δ5 desaturation) and the “Δ8 pathway” (elongation–Δ8 desaturation–Δ5 desaturation). DHA is achieved downstream in the biosynthesis of LC-PUFA from EPA via two alternative routes. Either, two consecutive elongations of EPA produce tetracosapentaenoic acid (TPA, 24:5n-3), which then undergoes a Δ6 desaturation to tetracosahexaenoic acid (THA, 24:6n-3), the latter being β-oxidised to DHA in peroxisome organelles following the translocation from endoplasmic reticulum, the so-called “Sprecher pathway” identified in mammals [24,31], or, the direct Δ4 desaturation of docosapentaenoic acid (DPA, 22:5n-3) into DHA via the ’Δ4 route´. The first Fads2 gene with Δ4 activity was identified in the marine herbivorous fish *Siganus canaliculatus* [32]. It was not only the first enzyme with this activity among fish, but it was the first discovered case in all vertebrates. The discovery indicated that there exists another possible mechanism for DHA biosynthesis, a direct route involving elongation of EPA to 22:5n-3 followed by Δ4 desaturation. If both DHA routes were coexist, this would represent a clear advantage for satisfying DHA requirements through endogenous production. After further identification of 11 teleost species having a putative Δ4 Fads2 by Oboh et al. [33], it was made clear that the direct Δ4 pathway is more widespread among teleost fish than initially believed.

## 4. Fads Gene Repertoire in Fish

There is a fundamental difference between fish and mammals regarding the gene repertoire encoding for enzymes performing desaturation activities needed for LC-PUFA biomolecules production. In contrast to mammals, where distinct separate genes *Fads1* and *Fads2* encode enzymes Fads1 and Fads2 with appropriate specificities Δ5 and Δ6 [34], respectively, in fish, *Fads1* gene has been lost during the evolution. As a result, all desaturation steps of the LC-PUFA biosynthetic pathway in fish are catalysed by Fads2 enzymes, exhibiting different Δ activities which can be overlapping to some extent.

Until recently, this scenario was generally accepted with no exceptions. Accordingly, one single Δ6 Fads2 appears most often [35,36,37,38]. Less often, a separate Δ6 Fads2 and Δ5 Fads2 paralogues appeared, such as in Atlantic salmon (*Salmo salar)* [39,40], or more than one single Δ6 Fads2 paralog are present such as in common carp (*Cyprinus carpio*) [41] and recently confirmed in numerous Osteoglossomorpha species [42]. In some teleosts studied, Δ6 Fads2 had measurable levels of Δ5 activity [37] or Δ8 activity [42,43]. A single bifunctional Δ6/Δ5 Fads2 acts in zebrafish (*Danio rerio*) [44] which was the first functionally characterized fish desaturase and for some time, it has been considered as an exception, not only in fish but in vertebrates in general. Later, two desaturases from marine rabbitfish (*Siganus canaliculatus*) were functionally characterized, one of which was shown to be Δ6/Δ5 bifunctional and the other Δ5/Δ4 bifunctional [32]. There exist extreme exceptions as well, represented by teleosts lacking *Fads*-like genes in their genomes, namely pufferfish *Takifugu rubripes* and *Tetraodon nigroviridis* [42]. The Atlantic salmon Δ6 and Δ5 *Fads2* cDNAs are very similar, sharing greater than 95% nucleic acid identity, indicating the presence of a recently duplicated locus, probably as the result of the recent salmonid whole genome duplication event [40].

The property of fish Fads2 exhibiting a more varied spectrum of Δ activities towards substrates has been hypothesized by Castro et al. [45] as a result of a functionalization process that occurred in response to dietary availability in natural pray. Functionally characterized Fads2 in numerous teleosts and all their activities determined by heterologous expression in yeast are listed in recent review of Kabeya et al. [46]. However, the persisting lack of information in some teleost lineages, such as Elopomorpha, and other nonteleost lineages, such as Lepisosteiformes, Polypteriformes or Cyclosomata, has hampered the full comprehension of Fads enzymes function in fish for a long time. Current novel insights into the fish LC-PUFA biosynthesis have provided a study on Fads desaturases published by Lopes-Marques et al. [42]. Accordingly, two types of desaturase repertoire are confirmed to appear in teleost fish, separating Elopomorpha from the other living teleost lineages. The orthologous gene to *Fads1* has been found in Japanese eel (*Anguilla* japonica), an Elopomorpha teleost specie, and confirmed by heterologous expression approach in yeast that desaturates the corresponding fatty acid substrates in the Δ5 position as well as sharing the common structural features to mammalian Fads1 enzymes. Farther Fads1 have been identified in some representatives of ancient fish lineages such as the Senegal bichir (*Polypterus senegalus*) and spotted gar (*Lepisosteus oculatus*) by these authors [42].

Based on sequence and phylogenetic data, *Fads2* and *Fads1* genes have been deduced to originate from the vertebrate ancestor and *Fads1* seems to be lost in Teleostei lineages except in Elopomorpha. It could be hypothesized that some teleosts have generated a mechanism to overcome the bottleneck caused by the loss of Δ5 Fads1, since otherwise, they would not be able to convert PUFA to LC-PUFA. Such a mechanism would be *Fads2* gene duplication followed by the process of functionalization as most probably was the case in salmonids, whereby acquisition of Δ5 *Fads2* occurred in one of the several *Fads2* gene copies. Another example would be the zebrafish (*Danio rerio*) in which Δ6 Fads2 acquired the ability to desaturate even in the Δ5 position [42,44]. The loss of canonical *Fads1* gene followed by *Fads2* subfunctionalization that teleosts have undergone during evolution could be linked to and explain the higher plasticity with which fish produce LC-PUFA biomolecules in comparison to other vertebrates.

## 5. Fads2 Structure and Structural Implications of Substrate Specificity

Fads2 are modular proteins which characteristically have a cytochrome *b5*-like domain on the N-terminus and the main desaturation domain with three histidine-rich regions on the C-terminus [47,48,49]. The fusion of the cytochrome *b5*-like domain to the main desaturase protein domain enables the NADH cytochrome *b5* reductase to directly transfer electrons to the catalytic site of Fads2 via the cytochrome *b5*-like domain without the requirement for an independent cytochrome *b5* [50,51]. However, solid evidence has been provided that both the cytochrome *b5*-like domain of Fads2 and microsomal cytochrome *b5* are necessary in the process of Δ6 desaturation and that the microsomal cytochrome *b5* does not compensate for the role of cytochrome *b5*-like domain of Fads2, which is accompanied by highly conserved heme-binding HPGG motif of the cytochrome *b5*-like domain. Moreover, protein–protein interactions between Fads2 and microsomal cytochrome *b5* are required for proper Fads2 function [52]. Phylogenetic studies have shown that cytochrome *b5* domain from Δ6 Fads2 proteins form a single cluster which points to a single ancient fusion event that took place in the common ancestor of all eukaryotes [53].

As a hydrophobic membrane-bound protein, Fads2 is extremely recalcitrant to characterization by conventional biochemical methods. A three-dimensional structure of Fads2 by X-ray crystallography is missing to date. The only animal desaturase whose structure is known is the stearoyl-CoA desaturase with Δ9 desaturation activity for which crystal structures have been published in humans [54] and rats [55]. There are some characteristic features common to all desaturases. The amino acid sequences within the substrate binding channel in all membrane desaturases contain the three His-boxes which histidine residues hold two irons in the active site. These histidine residues are of high evolutional conservation [56,57] and take place in very close proximity to the fatty acid substrate, referred to as “contact residues” [51]. Hydropathy analyses have shown that desaturases contain up to three long hydrophobic domains which are long enough to span the membrane bilayer twice whereas the His-boxes have a consistent positioning with respect to these potential membrane spanning domains. Sayanova et al. [56] undertook a massive motif analysis in more than fifty eukaryotic genomes, obtaining 275 desaturases, and reported the sequence logo representations of conserved histidine regions shown in Figure 2.

The first report on the structural basis of the substrate specificity of a mammalian front-end fatty acid desaturase was published by Watanabe et al. in 2016 [58]. Using the crystal structure modelling of the human soluble stearoyl-CoA (Δ9) desaturase [54,55], these authors performed homology modelling and revealed that Arg216, Trp244, Gln245, and Leu323 are located near the substrate-binding site. They applied site-directed mutagenesis to create mutations in rat Δ6 Fads2 at those sites they had predicted to influence the enzymatic function. Then, they exchanged these amino acids accordingly to be the same as in the unique bifunctional Δ6/Δ5 Fads2 from zebrafish. They determined amino acid residues responsible for both switching and adding the substrate specificity of rat Δ6 Fads2. Additionally, they predicted tertiary structure of rat Δ6 Fads2. There is a very important outcome from their publication—that one single amino acid in the Δ6 Fads2 desaturase enzyme, when changed, has the potential to switch the specificity towards substrates.

This corresponded to results from investigations on sex pheromones in moths [59], where similarly, a change as small as a single amino acid substitution in a fatty acid desaturase enzyme was sufficient to change the enzymatic function of the whole enzyme, moreover, resulting in huge consequences in reproduction. According to their data delivered, MsexD2 desaturase gene in *Manduca sexta* duplicated during the evolution whereby one copy acquired one amino acid change. Then, in the process of neofunctionalization, this novel gene acquired the ability to introduce another double bond and produce an uncommon sex pheromone with significant implication in species reproduction.

Corresponding data were obtained by the study of Δ6 Fads2 from marine algae *Thalassiosira pseudonana* [60]. Mutation sites in Δ6 Fads2 from *T. pseudonana* were determined which appeared to induce a propensity for the enzyme to favor binding of a particular fatty acid, suggesting that these may be associated with substrate specificity. The focused primarily on desaturation kinetics and assessed molecular mechanisms underlying the catalytic activity of Fads2 in *T. pseudonana*, because this model organism offers the advantage of exhibiting a very high desaturase catalytic activity suitable for such studies. They divided the amino acid sequence of Fads2 into sections and at the same time, they have used Fads2 from *Glossomastix chrysoplasta*, which in the opposite has very low enzymatic activity, and divided it in the same sections. To determine the catalytic activity of each region, the corresponding regions of both Fads2 enzymes were systematically exchanged to construct recombinant swap genes, which were expressed in yeast. Kinetics of enzymatic catalytic activity of recombinant desaturase were measured as well as amino acid residues important for catalytic activity by the use of site-directed mutagenesis were determined. As a result, topology prediction was created depictured in Figure 3. Amino acid substitutions significantly impacted the desaturation catalytic efficiency providing a solid basis for in-depth understanding of catalytic efficiency of Δ6 Fads2 enzyme.

The abovementioned studies clearly demonstrated that the strictness of structure-function relationship of Fads2 enzymes might be enormous and acquired changes as small as one single amino acid of enzyme primary structure might have significant consequences for the organism studied which could be a general feature extrapolatable even to more diverse taxa such as fish. Comparative studies of highly effective and minimally effective LC-PUFA biosynthetic machineries either between more or less related species or occurring in one single species (as typically studied in salmonids [29,40,61] have justified, that this is a promising strategy with great potential to gain insight into the challenging LC-PUFA biomolecules research in fish.

The question why some species can survive in EPA and DHA poor environment and other even closely related species not, has been an attractive research topic in the very recent past. An interesting structure-function study performed by Xie et al. [62] has addressed that question by studying the *Fads2* promoter sequence. The binding site for stimulatory protein Sp1 has been found as lacking in the promoter of *Fads2* gene in marine teleost *Epinephelus coioides*. The authors speculated therefore, that the Sp1-binding site absence might be the main cause of the very low Fads2 expression in marine carnivorous teleost species. To test this hypothesis, they inserted the Sp1-binding site from the *Fads2* promoter sequence of the herbivorous *Siganus canaliculatus*, the first marine teleost demonstrated to have LC-PUFA biosynthetic ability, into the corresponding region of *E. coioides Fads2* promoter sequence. As expected, Δ6 desaturation activity increased significantly. The results provided direct evidence for the importance of the Sp1-binding site in determining *Fads2* promoter activity and indicated that its lack may be a reason for very low expression of Fads2 and poor LC-PUFA biosynthetic ability in *E. coioides*. The Sp1-binding site has been found as lacking in marine carnivorous fish *Gadus morhua* [36] as well as in *Dicentrarchus labrax* [63], while in *Oncorhynchus mykiss* its promoter activity was weaker [64].

## 6. Fads2 Copy Number Variation

The best was yet to come regarding studying marine vs. freshwater fish dealing with LC-PUFA poor food sources. Just a few months ago, Science released an exciting paper from Ishikawa et al. [65]. In this comprehensive study, the authors compared three-spined stickleback (*Gasterosteus aculeatus* species complex) which successfully colonized newly emerged freshwater bodies after glacial retreat with closely related marine Japan Sea stickleback (*G. nipponicus*) which had failed to colonize freshwater. They linked the colonization success to *Fads2* gene copy number, being higher in Pacific Ocean stickleback from the *G. acuelatus* complex. When transgenic Japan Sea stickleback overexpressing Fads2 was made and fed only DHA-free *Artemia,* the Fads2 transgenics showed a higher survival rate and higher DHA content at 40 days after fertilization than the control GFP-transgenics. Moreover, *Fads2* gene linkage to X chromosome was confirmed, resulting in higher copy number in females. That fact is consistent with higher female survival observed. These results suggested that lower *Fads2* copy number may be a constraint to colonization of DHA-deficient freshwater niches by Japan Sea stickleback. These authors went deeper into the copy number assessing in the context of phylogeny and deduced a general mechanism: Higher *Fads2* copy number contributes to survival with DHA-free diets. Hence, *Fads2* was the metabolic gene important for overcoming the nutritional constraints associated with freshwater colonization in fishes. The authors mentioned the intriguing feature of the *Fads2* gene to make strong signatures of selection even in human such as in Greenland Eskimos which might be farther extrapolated to even more diverse taxa.

## 7. Fads2 Transgenes

The limited availability of LC-PUFA derived from fish represents the critical bottleneck in food production systems, one that numerous research institutions and aquafeed companies in this field are trying to overcome. Attempts to replace fish-derived LC-PUFA by plant derived alternatives often resulted in low quality products lacking the original content of these health promoting biomolecules [66]. This problem could be minimized by either feeding fish genetically modified plants for enhanced EPA and DHA production or by gene editing fish to be capable to produce endogenous LC-PUFA more effectively. Genetic engineering has been long been utilized as a strategy to increase natural productivity. Genetically engineered organisms could have the potential to reduce pressure on current LC-PUFA natural resources. The efforts and progress to develop transgenic plants as terrestrial sources of ω3 fish oils as well as advances in the field have been reviewed recently by Napier [67]. Transgenic fish have many potential applications in aquaculture, but the research also raised concerns regarding the possible risks to the environment associated with release and escape. A tabulated balance sheet of likely benefits and risks have been published by Maclean and Laight [68]. In this review, we focused on attempts to produce genetically modified fish with an enhanced content of ω3 LC-PUFA.

The first step to modifying the LC-PUFA biomolecules production pathway using genetic engineering was done in zebrafish [69], into which a gene for Δ5 Fads2 from masu salmon was introduced. The result demonstrated that masu salmon (*Oncorhynchus masou)* Δ5 Fads2 is functional in zebrafish and modifies its LC-PUFA metabolic pathway; hence, the technique could be applied to farmed fish to generate a nutritionally richer product for human consumption. The closely relative to zebrafish, the common carp (*Cyprinus carpio*) accounts for about 40% of the total global aquaculture production and could therefore deliver a significant amount of LC-PUFA if they were produced in their body. However, the content of EPA and DHA (mg g^−1^) in muscle tissue of common carp is relatively low when compared to many other fish species, as revealed by recent meta-analysis data [70].

Some pioneering transgenesis experiments were carried out which reported trends towards increased ω3 LC-PUFA content in muscle of transgenic progeny—Δ5 *Fads2* from masou salmon driven by a β-actin promoter was introduced into common carp [71] and channel catfish (*Ictalurus punctatus*) [72] with the aim to improve ω3 LC-PUFA production. The results have shown promise for future work in this area, when utilizing homozygous transgenic individuals in contrast to the heterozygous individuals utilized in these studies. The effects of the transgene varied between common carp and channel catfish, being higher in common carp [72].

However, only a few month ago, ´*Haiyouli´* construction was published [73], which, in Chinese, means “advantageous carp-like marine fish”. This common carp was genetically modified with the aim to elevate production of ω3 LC-PUFA. The transgene used was a fish-codon optimized fatty acid desaturase (*fat1*) coding sequence originally from *Caenorhabditis elegans* driven by the 5´upstream regulatory region of common carp β-actin. Unexpectedly for the authors, under transgene expression fat accumulation of the internal organs decreased and in the liver tissues, *fat1*-transgenic common carp showed less accumulation of lipid droplets when compared with wildtype. However, the quantitative RT-PCR results showed a 10.5-fold increase in Fads2 expression, a 6.5-fold increase in elongase 5 expression and a 3-fold increase in elongase 2 expression in the transgenic tissue, indicating stimulation of LC-PUFA biosynthesis by the expression of exogenous *fat1* desaturase. Interestingly, the transcription of acyl-CoA oxidase 3 increased by 8.2 in transgenic tissue, which perfectly explains the lipid content decrease in internal organs of genetically modified common carp. Intriguingly, the authors stated that the ω6 to ω3 ratio of their transgenic common carp (0.4) was even lower than that of the Atlantic salmon (0.58) reported by Henderson and Tocher [74]. For this reason, *Haiyouli* has been presented as a potentially ideal fish produced in modern society to balance the high ω6 to ω3 ratio of human diets. However, such a conclusion is questionable since they stimulate LC-PUFA biosynthesis by the expression of exogenous *fat1* at the expense of overall lipid biosynthesis. When total mass unit of LC-PUFA per mass unit of filet is calculated and compared to salmon, there may, in fact, be no relative advantage to consuming *Haiyouli*. Hence, it is rather a step forward on the way to constructing the ideal fish.

Successful production of DHA using *Fads2* transgenes has been reported in mammals. In Chinese hamster ovary cells, LC-PUFA-elevated production was achieved by heterologous expression of fish Δ4 Fads2 from *Siganus canaliculatus* with concomitant overexpression of Δ6 Fads2 and Δ5 Fads1 from mice. The authors stated that this new technology has been confirmed as very effective in high-level production of DHA from dietary ALA and provided a potential for the creation of new land animal breeds who could produce DHA abundantly in their related products [75].

If such solutions come into practice, this will have a positive effect in sufficient delivery of health promoting LC-PUFA to humans while at the same time, preserving wild fish populations.

## 8. Conclusions

Fads2 is a fascinating enzyme with far-reaching implications for both human health and environmental sustainability. It is clear that Fads2 has played an important role in the adaptations to novel environments throughout evolutionary history as differences in both gene expression and copy number have been reported across freshwater and seawater dwelling species. We have demonstrated the importance of this enzyme in the context of growing aquaculture demand and dwindling LC-PUFA supply and outlined a few remedies in the form of genetic engineering to improve endogenous PUFA production. By improving our understanding of Fads2, we can address major environmental concerns and break out of the cycle of exploitation that currently strains our wild fish reserves to feed the growing aquaculture sector.

## Figures and Tables

**Figure 1 biomolecules-10-00206-f001:**
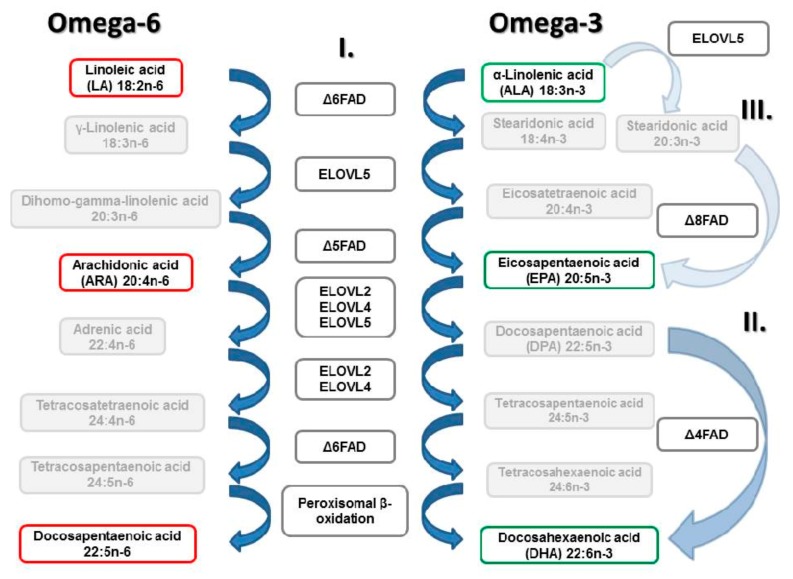
Biosynthetic pathways of long-chain (≥ C_20_) polyunsaturated fatty acids (LC-PUFAs) of ω3 and ω6 families from dietary essential α-linolenic (ALA, 18:3n-3) and linoleic (LA, 18:2n-6) acids, respectively, by elongation and desaturation reactions. Adapted from (Carmona-Antoñanzas et al., 2011; Monroig et al., 2011; Sprecher, 2000; Voss et al., 1991); modified after (Trattner, 2009; Vestergeren, 2014; Yan, 2016).

**Figure 2 biomolecules-10-00206-f002:**
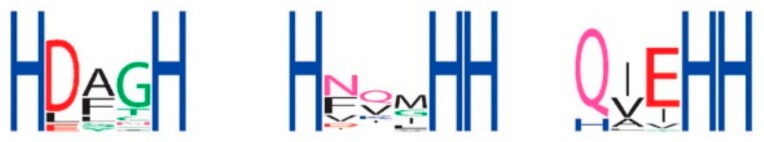
Sequence logo of histidine boxes in membrane associated front-end desaturases such as Δ6 Fads2. The high of letters corresponds to the occurrence probability. Adapted from (Sayanova, 2001). Modified after (Hashimoto, 2007).

**Figure 3 biomolecules-10-00206-f003:**
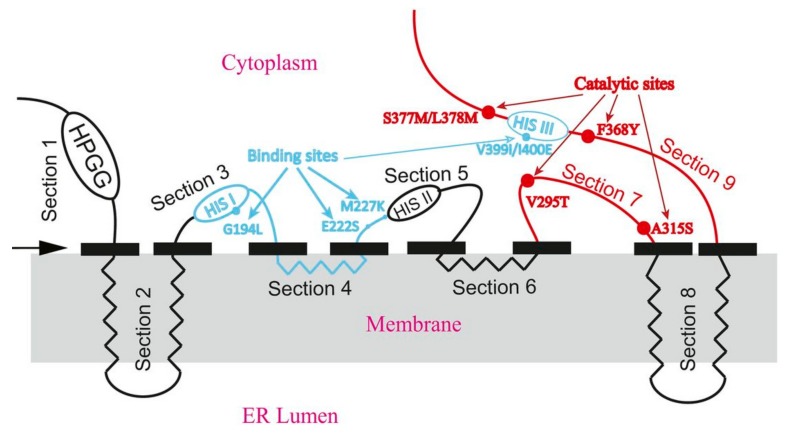
The predicted topology model of Δ6 Fads2. The black solid rectangles indicate boundaries of domains. Four alpha-helices span the membrane. The blue lines and blue dots indicate the areas and sites implicated in substrate specificity, red lines and red dots indicate the areas and sites important for catalytic activity, respectively. ER lumen: endoplasmic reticulum lumen. HIS I, HIS II and HIS III: histidine rich motifs. Modified from (Shi et al., 2018).

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
