# Peer review of "Assessment of Fatty Acid Desaturase (Fads2) Structure-Function Properties in Fish in the Context of Environmental Adaptations and as a Target for Genetic Engineering"

_biomolecules, 2020, doi:10.3390/biom10020206_

Round 1

Reviewer 1 Report

The authors did much work to amend the manuscript according to the previous comments. The review would be a valuable issue after some more revisions in sense and style of the text.

I suggest to correct the title like following: Assessment of fatty acyl desaturase (Fads2) structure-function properties in fish in the context of environmental adaptations and as a target for genetic engineering

Line 18-20 “Endogenous production” is mixed here for vertebrates, fish, and humans. There in no aim to increase “endogenous production” of LC-PUFA in humans or all vertebrates, this objective is for fish. The problem must be described here more accurate and narrow. 

Line 53 Insects are also animals. Specify, please, what animals are considered here.

Line 66 Use humans instead of human.

Line 76-77 Add the names of enzymes that “catalysing subsequent steps in LC-PUFA biosynthesis” here. I guess that Fads2 makes al least two steps in LC-PUFA biosynthesis. If you formulate in this way, you should at least cite papers showed importance of the other enzymes, e.g., elongases.

Line 88 “The essentiality of fat as an essential dietary component” is a strange expression with ambiguous meaning. It is better to use lipids instead of fat.

Line 109-114 The new data reviewed in the paper by Gladyshev & Sushchik, 2019 do not contradict to the necessity of balanced abundance of EFA and other lipids in diet. They showed various strategies of human populations regarding EFA metabolism. 

Line 161 Are fads1 and fads2 the designations for desaturation genes? If so, give them in the same manner in all the text, i.e., either in normal or in italic font. Both variants now present in the text. Check and correct, if necessary, the text also for the uniform designation of the desaturase enzymes.

Line 161 A gene can not be “usually lost during an evolution”. Edit the phrase.  

Line 202-213 The information presented in the paragraph is hardly related to the main topic of this section. I suggest to remove it or shorten and highlight an importance of cytochrome b5 domain structure for regulation of the enzyme substrate specificity.

Line 246-250 Condense or shorten the data regarding insect sex pheromones.

Line 329 Meaning of the phrase “to produce healthy LC-PUFA in fish” is unclear. Edit the phrase.

Line 334 Give Cyprinus carpio as italic font.

Line 336-337 Insert, please, “common carp” after “muscle tissue”.

Line 340 Give masu as lower-case.

Line 342-345 Meaning of the phrase is unclear. Two studies were mentioned in the previous sentence; why here is only one? Did the cited papers deal with content of n-3 LC-PUFA expressed as mg per g of mass (or similar units)? If only the n-3 LC-PUFA percentages of the total FA were considered, it is too early to emphasize positive effects of the transgenes.

Line 347-360 There is no “substantially elevated production of ω3 LC-PUFA” in this genetically modified common carp. In fact, the cited authors succeeded to stimulate “LC-PUFA biosynthesis by the expression of exogenous fat1 desaturase” at the expense of the decrease in overall lipid biosynthesis. You correctly describe all molecular and biochemical findings of the cited paper. However, the final conclusion of this paper that “Haiyouli as a potentially ideal fish” is wrong, since it bases on percentages of LC-PUFA, rather than their contents as mg per g of mass. If to calculate and compare LC-PUFA content per mass unit (on the base of total lipid content and LC-PUFA percentages) between control and transgenic common carp lines, there would be no advantage of the transgenic line. In the studied transgenic common carp, total lipid content per mass unit of filet strongly decreased, and even the achieved increase of LC-PUFA biosynthesis could not fully compensate decrease in LC-PUFA content that was related with decrease in overall lipid accumulation.

Author Response

Dear reviewer 1,

We are very grateful for Your effort and time to help us by providing valuable comments and suggestions. We have improved our review accordingly.

Comments and Suggestions:

The authors did much work to amend the manuscript according to the previous comments. The review would be a valuable issue after some more revisions in sense and style of the text.

I suggest to correct the title like following: Assessment of fatty acyl desaturase (Fads2) structure-function properties in fish in the context of environmental adaptations and as a target for genetic engineering.

We have corrected the title as suggested. However instead of “acyl” we have used “acid” consistently to the text of the review.

Line 18-20 “Endogenous production” is mixed here for vertebrates, fish, and humans. There in no aim to increase “endogenous production” of LC-PUFA in humans or all vertebrates, this objective is for fish. The problem must be described here more accurate and narrow. 

Specified as production species.

Line 53 Insects are also animals. Specify, please, what animals are considered here.

            We have removed “(insects)” as the paper cited states that insects are also dependend on their diet to obtain ω3 and ω6 fatty acids. The only exception appears to be the cockroach which has a delta 12 desaturase and is therefore nutritionally independent of dietary lipid and to synthesize eicosanoids de novo. But we do not feel to go to such a deep detail in the introduction. Hence, we introduced the term “usually” dependent on plants.

Line 66 Use humans instead of human.

corrected

Line 76-77 Add the names of enzymes that “catalysing subsequent steps in LC-PUFA biosynthesis” here. I guess that Fads2 makes al least two steps in LC-PUFA biosynthesis. If you formulate in this way, you should at least cite papers showed importance of the other enzymes, e.g., elongases.

We have added elongases Elovl4 and Elovl5

Line 88 “The essentiality of fat as an essential dietary component” is a strange expression with ambiguous meaning. It is better to use lipids instead of fat.

corrected

Line 109-114 The new data reviewed in the paper by Gladyshev & Sushchik, 2019 do not contradict to the necessity of balanced abundance of EFA and other lipids in diet. They showed various strategies of human populations regarding EFA metabolism. 

Thank you for this comment, we corrected the text accordingly.

Line 161 Are fads1 and fads2 the designations for desaturation genes? If so, give them in the same manner in all the text, i.e., either in normal or in italic font. Both variants now present in the text. Check and correct, if necessary, the text also for the uniform designation of the desaturase enzymes.

corrected

Line 161 A gene can not be “usually lost during an evolution”. Edit the phrase.

The term “usually” removed.

Line 202-213 The information presented in the paragraph is hardly related to the main topic of this section. I suggest to remove it or shorten and highlight an importance of cytochrome b5 domain structure for regulation of the enzyme substrate specificity.

We have changed the topic of the section as “Fads2 structure and structural implication of substrate specificity” instead of “ Structural implication of Fads2 substrate specificity”.

Line 246-250 Condense or shorten the data regarding insect sex pheromones.

shortened slightly

Line 329 Meaning of the phrase “to produce healthy LC-PUFA in fish” is unclear. Edit the phrase.

edited

Line 334 Give Cyprinus carpio as italic font.

corrected

Line 336-337 Insert, please, “common carp” after “muscle tissue”.

inserted

Line 340 Give masu as lower-case.

given

Line 342-345 Meaning of the phrase is unclear. Two studies were mentioned in the previous sentence; why here is only one? Did the cited papers deal with content of n-3 LC-PUFA expressed as mg per g of mass (or similar units)? If only the n-3 LC-PUFA percentages of the total FA were considered, it is too early to emphasize positive effects of the transgenes.

We absolutely agree that it is too early to emphasize positive effects of the transgenes. For this reason we cite these two works very carefully as pioneering experiments which report trends towards increased ω3 LC-PUFA which could be promising in the future under the condition of utilizing homozygous instead of heterozygous individuals. To give the best answer to your questions, allow us to cite the W. Bugg´s work:

            “Quantification of n-3 fatty acid levels was accomplished through gas chromatography-mass spectrometry. In general, n-3 fatty acid production was higher in transgenic individuals than in controls. Channel catfish showed no significant difference in n-3 fatty acid production because of low sample size, but trends towards increased production can be seen for n-3 fatty acids, in terms of total FAME percentage, α-linolenic acid (ALA) and docosahexaenoic acid (DHA), however these were not significantly different from controls (P = 0.212 and 0.207 respectively). The product on the n-6 side, arachidonic acid (AA), increased by 12.86% (P = 0.8). Precursors to n-6 delta5-desaturation, linoleic acid (LA) and dihomo-γ- linoleic acid (DGA) decreased 13.2% and 11.87% respectively (P = 0.116 and 0.8 respectively). On a mg/g weight basis for eicosapentaenoic acid (EPA) (P = 0.089), DHA (P = 0.078), total HUFA (P = 0.056) were higher in transgenic channel catfish. Fatty acid profiles for transgenic channel catfish were significantly more uniform compared to their control counter parts. Transgenic F1 common carp showed variation in lipid profile and higher level of saturated fatty acids when compared to controls. Transgenic F1 common carp showed higher variation in lipid profile and higher level of saturated fatty acids when compared to controls. Total fatty acid production of F1 desaturase common carp (9.55mg) and controls (9.93mg) per gram wet weight of muscle were not different (P = 0.68). As a percentage of FAME, desaturase F1 common carp showed a 1.14-fold decrease (P = 0.35) in n-3 fatty acid levels measured as a change in ALA, DHA, and EPA compared to controls. Individually, as a percentage of total fatty acids, ALA increased 20.72% (P = 0.18).

Desaturase transgenesis affected n-3 fatty acid production in a positive direction, showing promise for future work in this area, and when utilizing homozygous transgenic individuals in contrast to the heterozygous individuals utilized in the current study.”

Line 347-360 There is no “substantially elevated production of ω3 LC-PUFA” in this genetically modified common carp. In fact, the cited authors succeeded to stimulate “LC-PUFA biosynthesis by the expression of exogenous fat1 desaturase” at the expense of the decrease in overall lipid biosynthesis. You correctly describe all molecular and biochemical findings of the cited paper. However, the final conclusion of this paper that “Haiyouli as a potentially ideal fish” is wrong, since it bases on percentages of LC-PUFA, rather than their contents as mg per g of mass. If to calculate and compare LC-PUFA content per mass unit (on the base of total lipid content and LC-PUFA percentages) between control and transgenic common carp lines, there would be no advantage of the transgenic line. In the studied transgenic common carp, total lipid content per mass unit of filet strongly decreased, and even the achieved increase of LC-PUFA biosynthesis could not fully compensate decrease in LC-PUFA content that was related with decrease in overall lipid accumulation.

Thank you for this comment and help, we have adjusted lines 346-363 accordingly and highlighted the questionability of the conclusion of “Haiyouli” authors.

Yours sincerely.

Zuzana Bláhová

Reviewer 2 Report

Letter to Authors,
biomolecules-701596-v1
Assessment of Fads2 products and structure-function properties in the context of environmental adaptations and as a target of fish genetic engineers
Zuzana Blahova, Thomas Nelson Harvey, Martin Psenicka, Jan Mraz

200122

Dear authors,
Your review paper well documents roles and importance of Fads2 in LC-PUFA production in animals, especially in fish. Because considerable proportion of LC-PUFA in animal bodies comes from food webs, and because anthropogenic activities threat primary production of LC-PUFA in ecosystems, there will be growing needs for studies on animal enzymes involved in endogenous LC-PUFA production. This review paper deals thus with very timely subjects, and will have impacts on aquaculture science and industry.
Before publication, you should consider revising your manuscript to make it clear, direct, strong and straightforward. Use abbreviations consistently. You should also address the carp issue on aquatic environments.
See below for detail.

L32 keywords
Replace fads2 with another word that do not appear in the title to draw attention from wider readership.

L33
EFA (not very common) -> essential fatty acid
See L93.

L40
less unsaturated PUFA -> lower level polyunsaturated fatty acid (PUFA)
Spell-out at the first place independent from the abstract. Avoid repetition.

L52
linoleic acid -> LA
alpha-linolenic acid -> ALA
Use abbreviations consistently. See L41.

L59
are aquatic ecosystems (supply are) -> is [ultimately] from aquatic ecosystems

L66
eutrophication -> eutrophication, (insert a comma)

L69
could be seen -> it could be seen ?

L88
essentiality (avoid repetition) -> {importance, implication, pivotal role, [delete]}

L94
But (do not begin with 'but') -> connect sentences

L101
But -> However,

L110
technologies -> technologies, (insert a comma)

L129
docosahexaenoic acid (DHA) -> DHA
See L39.

L130
Arachidonic acid (AA) and eicosapentaenoic acid (EPA) -> AA and EPA
See L39,40

L148
An embedded text tells this is Figure 3.

L159-199
Long paragraph. Break at L164,177,192.

L184
last year -> delete

L192
instead of -> except

L205,206
b5 -> in Italics?

L252,311,358,359
The authors -> They ?
Readers will be embarrassed to see which 'the authors' points.

L338-339
This argument is unacceptable. You should address some more about potential conflicts between environmental protection and carp farming. The common carp is a well-known ecosystem engineer and one of the World's worst invasive alien species. Nuisance of overpopulated carps has been documented even in its native range.

L342,345
this study ?
You cited two studies. Do you mean 'this' is your present paper?

L353,354,356,358,
e.g. 10,5 -> 10.5
Use a dot instead of comma for decimal point.

L392 references
Check the reference list carefully again from the beginning. Reference lists are frequently den of errors. Does citation in the text correspond with the list? Did literature numbering shift on the way internal revision? It is the authors' responsibility that all references are properly cited.

The following items may be helpful for further discussion.

IUCN Invasive Species Specialist Group. 2013. 100 of the World's Worst Invasive Alien Species. http://www.iucngisd.org/gisd/100_worst.php

Matsuzaki SS, Usio N, Takamura N, Washitani I. 2009. Contrasting impacts of invasive engineers on freshwater ecosystems: an experiment and meta-analysis. Oecologia 158:673-686.

Rahman MM, Verdegem MCJ, Nagelkerke LAJ, Wahab MA, Milstein A, Verreth JAJ. 2008. Effects of common carp Cyprinus carpio (L.) and feed addition in rohu Labeo rohita (Hamilton) ponds on nutrient partitioning among fish, plankton and benthos. Aquacult Res 39:85-95.

Stroud CK, Nara TY, Roqueta-Rivera M, Radlowski EC, Lawrence P, Zhang Y, Cho BH, Segre M, Hess RA, Brenna JT, Haschek WM, Nakamura MT. 2009. Disruption of FADS2 gene in mice impairs male reproduction and causes dermal and intestinal ulceration. J Lipid Res 50:1870-1880.

Stuart IG, Jones MJ. 2006. Movement of common carp, Cyprinus carpio, in a regulated lowland Australian river: implications for management. Fish Manag Ecol 13:213-219.

Author Response

Dear reviewer 2,

We are very grateful for Your effort and time to help us by providing valuable comments and suggestions. We have improved our review accordingly.

Comments and Suggestions:

Dear authors,
Your review paper well documents roles and importance of Fads2 in LC-PUFA production in animals, especially in fish. Because considerable proportion of LC-PUFA in animal bodies comes from food webs, and because anthropogenic activities threat primary production of LC-PUFA in ecosystems, there will be growing needs for studies on animal enzymes involved in endogenous LC-PUFA production. This review paper deals thus with very timely subjects, and will have impacts on aquaculture science and industry.
Before publication, you should consider revising your manuscript to make it clear, direct, strong and straightforward. Use abbreviations consistently. You should also address the carp issue on aquatic environments.
See below for detail.

L32 keywords
Replace fads2 with another word that do not appear in the title to draw attention from wider readership.

replaced

L33
EFA (not very common) -> essential fatty acid
See L93.

changed

L40
less unsaturated PUFA -> lower level polyunsaturated fatty acid (PUFA)
Spell-out at the first place independent from the abstract. Avoid repetition.

replaced

L52
linoleic acid -> LA
alpha-linolenic acid -> ALA
Use abbreviations consistently. See L41.

used

L59
are aquatic ecosystems (supply are) -> is [ultimately] from aquatic ecosystems

changed

L66
eutrophication -> eutrophication, (insert a comma)

inserted

L69
could be seen -> it could be seen ?

added

L88
essentiality (avoid repetition) -> {importance, implication, pivotal role, [delete]}

used „pivotal role“

L94
But (do not begin with 'but') -> connect sentences

We apologize not to do that, because we intend to highlight the fact that the concept of essential fatty acids appeared before the reactions of desaturation and elongation had been demonstrated.

L101
But -> However,

replaced

L110
technologies -> technologies, (insert a comma)

inserted

L129
docosahexaenoic acid (DHA) -> DHA
See L39.

docosahexaenoic acid removed

L130
Arachidonic acid (AA) and eicosapentaenoic acid (EPA) -> AA and EPA
See L39,40

removed

L148
An embedded text tells this is Figure3.

L159-199
Long paragraph. Break at L164,177,192.

broken

L184
last year -> delete

deleted

L192
instead of -> except

            -changed

L205,206
b5 -> in Italics?

written in Italics

L252,311,358,359
The authors -> They ?
Readers will be embarrassed to see which 'the authors' points.

L252, L359 – rewritten

L338-339
This argument is unacceptable. You should address some more about potential conflicts between environmental protection and carp farming. The common carp is a well-known ecosystem engineer and one of the World's worst invasive alien species. Nuisance of overpopulated carps has been documented even in its native range.

Thank You for that comment. We have omited the sentence.

L342,345
this study ?
You cited two studies. Do you mean 'this' is your present paper?

We incorrectly have used „This“ for results given by the two papers. L340-344 – changed.

L353,354,356,358,
e.g. 10,5 -> 10.5
Use a dot instead of comma for decimal point.

corrected

L392 references
Check the reference list carefully again from the beginning. Reference lists are frequently den of errors. Does citation in the text correspond with the list? Did literature numbering shift on the way internal revision? It is the authors' responsibility that all references are properly cited.

We went through the reference list, corrected errors and added „doi“ where missing.

The following items may be helpful for further discussion.

Thank you for provided reference and apologise for not following that recommendations. We are afraid it would extend the review too much by taking the readers attention to other directions than we primarily intended.

IUCN Invasive Species Specialist Group. 2013. 100 of the World's Worst Invasive Alien Species. http://www.iucngisd.org/gisd/100_worst.php

Matsuzaki SS, Usio N, Takamura N, Washitani I. 2009. Contrasting impacts of invasive engineers on freshwater ecosystems: an experiment and meta-analysis. Oecologia 158:673-686.

Rahman MM, Verdegem MCJ, Nagelkerke LAJ, Wahab MA, Milstein A, Verreth JAJ. 2008. Effects of common carp Cyprinus carpio (L.) and feed addition in rohu Labeo rohita (Hamilton) ponds on nutrient partitioning among fish, plankton and benthos. Aquacult Res 39:85-95.

Stroud CK, Nara TY, Roqueta-Rivera M, Radlowski EC, Lawrence P, Zhang Y, Cho BH, Segre M, Hess RA, Brenna JT, Haschek WM, Nakamura MT. 2009. Disruption of FADS2 gene in mice impairs male reproduction and causes dermal and intestinal ulceration. J Lipid Res 50:1870-1880.

Stuart IG, Jones MJ. 2006. Movement of common carp, Cyprinus carpio, in a regulated lowland Australian river: implications for management. Fish Manag Ecol 13:213-219.

Yours sincerely.

Zuzana Bláhová